# Key Players and Biomarkers of the Adaptive Immune System in the Pathogenesis of Sarcoidosis

**DOI:** 10.3390/ijms21197398

**Published:** 2020-10-07

**Authors:** Emily-Rose Zhou, Sergio Arce

**Affiliations:** 1School of Medicine Greenville, University of South Carolina, Greenville, SC 29605, USA; ezhou@email.sc.edu; 2Department of Biomedical Sciences, School of Medicine Greenville, University of South Carolina, Greenville, SC 29605, USA

**Keywords:** sarcoidosis, granuloma, macrophages, T-cells, cytokines, chemokines, B cells, plasma cells, fibrosis

## Abstract

Sarcoidosis is a systemic inflammatory disease characterized by development of granulomas in the affected organs. Sarcoidosis is often a diagnosis of exclusion, and traditionally used tests for sarcoidosis demonstrate low sensitivity and specificity. We propose that accuracy of diagnosis can be improved if biomarkers of altered lymphocyte populations and levels of signaling molecules involved in disease pathogenesis are measured for patterns suggestive of sarcoidosis. These distinctive biomarkers can also be used to determine disease progression, predict prognosis, and make treatment decisions. Many subsets of T lymphocytes, including CD8^+^ T-cells and regulatory T-cells, have been shown to be dysfunctional in sarcoidosis, and the predominant CD4^+^ T helper cell subset in granulomas appears to be a strong indicator of disease phenotype and outcome. Studies of altered B cell populations, B cell signaling molecules, and immune complexes in sarcoidosis patients reveal promising biomarkers as well as possible explanations of disease etiology. Furthermore, examined biomarkers raise questions about new treatment methods and sarcoidosis antigens.

## 1. Introduction

Sarcoidosis is an inflammatory disease of unknown origin characterized by granuloma formation in the affected organs. It preferentially involves the lungs but may affect other organs systemically such as the skin, joints, and eyes. While other inflammatory diseases such as rheumatoid arthritis have specific biomarkers like the anti-cyclic citrullinated peptide (anti-CCP) and Rheumatoid Factor (RF) antibodies, sarcoidosis has no specific biomarker. Traditionally used markers of sarcoidosis such as Angiotensin Converting Enzyme (ACE) and lysozyme—enzymes produced by granuloma macrophages—display low sensitivity and/or specificity [1]. Although finding an antibody or a T-cell receptor (TCR) clonotype that binds an identified sarcoid antigen would produce an ideal biomarker, this goal has remained elusive until such an antigen and antibody/TCR clonotype are identified.

Additionally, more accurate methods of determining disease phenotype and progression are needed. The commonly used Scadding staging system based on chest x-ray phenotypes pulmonary sarcoidosis into five stages that generally increase with greater clinical pulmonary involvement [2]. Stage 0 is mild with no lung infiltrates or adenopathy; Stage 1 has only hilar lymphadenopathy; Stage 2 has hilar lymphadenopathy and lung infiltrates; Stage 3 has only lung infiltrates; and Stage 4 is the most severe with diffuse pulmonary fibrosis [2]. However, the Scadding system fails to consider extrapulmonary symptoms, which often have no correlation to chest x-ray findings [3]. The Scadding system also poorly correlates with other markers of disease severity such as pulmonary function tests, individual patient treatment needs, and disease severity reported by patients [3].

We propose an alternative way to screen for and stage sarcoidosis by examining signaling abnormalities and imbalances in B and T lymphocyte populations—i.e., new molecular and cellular biomarkers—involved in disease pathogenesis. Not only can these indicators be used to help diagnose disease and monitor disease progression, but they can potentially be used to detect more active or severe forms of sarcoidosis, predict the success of certain therapies, and explain disease etiology/mechanism (Figure 1).

## 2. T-Cell Response Biomarkers

It is generally accepted that sarcoidosis initially arises from the interaction between macrophages, Type 1 CD4^+^ T Helper (TH1) cells and antigens of unknown origin that drive the production of inflammatory cytokines such as Tumor Necrosis Factor (TNF)-α, Interleukin (IL)-12, IL-2 and Interferon (IFN)-γ which ultimately results in granuloma formation. Granulomas, the pathological hallmark of sarcoidosis, are tightly packed clusters of cells comprising a central core of macrophages, epithelioid histiocytes and multinucleated giant cells surrounded by a lymphocyte collar. The lymphocyte collar is composed mainly of CD4^+^ T-cells, but CD8^+^ T-cells, B cells, plasma cells and fibroblasts can also be found, though to a lesser extent [4]. CD4^+^ T lymphocytes are thought to play a critical role in sarcoidosis development by recruiting leukocytes, forming granulomas, and interacting with B cells to stimulate antibody production [2].

## 3. TH1/TH2 Cell Subtype Shifts and Detection of Sarcoidosis

Detectable immune cell population imbalances exist in sarcoidosis tissue. Immunohistochemistry (IHC) analysis of lung/mediastinal lymph node (MLN) biopsies from sarcoidosis patients show a skewed CD4:CD8 T-cell ratio of 4:1, similar to the ratio seen in tuberculoid granulomas, and bronchoalveolar lavage (BAL) fluid of sarcoidosis patients demonstrated marked lymphocytosis and an elevated CD4:CD8 ratio compared to BAL fluid of patients with different interstitial lung diseases (ILD) such as usual interstitial pneumonia, nonspecific interstitial pneumonia, and extrinsic allergic alveolitis [4,5,6]. Although recent meta-analysis has shown that elevated CD4/CD8 ratio lacks sufficient specificity (0.83) and sensitivity (0.70) to diagnose sarcoidosis, BAL CD4/CD8 analysis may still prove helpful in gauging likelihood of sarcoidosis versus other ILDs when combined with other measurements [7]. Additionally, peripheral blood of sarcoidosis patients had increased numbers of CD62L^+^ CD45RA^−^ CD45RO^+^ central memory and CD62^−^ CD45RA^−^ CD45RO^+^ effector memory CD4^+^ T-cells with significantly decreased numbers of CD62L^+^ CD45RA^+^ CD45RO^−^ naïve CD4^+^ T-cells [5]. The memory CD4^+^ T-cell population of sarcoidosis patients also contained a significantly increased frequency of CD27^−^ cells indicating chronic antigenic stimulation [5].

Examination of peripheral blood and granuloma CD4^+^ T-cells revealed altered cytokine production and multiple signal transduction dysfunctions. The inflammatory cytokines TNF-α, IFN-γ, Macrophage Inflammatory Protein (MIP)-1α, and IL-12 were all elevated in the BAL fluid of sarcoidosis patients, indicating a pro-inflammatory state as well as a shift toward the TH1 subtype of CD4^+^ T-cells and the classically activated M1 macrophages that TH1 cells promote [8]. High levels of CD40^+^ cells, a marker of M1 macrophages, in sarcoidosis BAL fluid further confirmed the prevalence of the M1 macrophage phenotype in these patients [9]. IFN-γ may exert its effect by inhibiting apoptosis of macrophages through expression of cyclin-dependent kinase inhibitor p21/Waf1 which was significantly increased in granulomas of sarcoidosis patients thus promoting persistent lung inflammation [10]. IFN-γ’s proposed anti-apoptotic effects are further supported by the observation that higher IFN-γ concentrations were also correlated with increased number of macrophages in BAL fluid of sarcoidosis patients [11]. In addition, IFN-γ and TNF-α promotes aggregation and fusion of macrophages into multinucleated giant cells characteristic of sarcoid granulomas [12].

Conversely, macrophages influence the predominant TH cell subtype present in the sarcoid lungs. C-X-C motif ligand 9, 10, and 11 (CXCL9, CXCL10 and CXCL11), are IFN-γ-induced chemokines that bind to C-X-C Chemokine Receptor Type 3 (CXCR3), a molecule preferentially expressed by TH1 cells [13]. CXCL9 and CXCL10 were significantly elevated in the BAL fluid of sarcoidosis patients and the levels of these two chemokines strongly correlated with increased numbers of effector TH1 cells in the inflamed lungs [13]. Alveolar macrophages, multinucleated giant cells, and epithelioid histiocytes in sarcoidosis lungs also secreted all three chemokines indicating active production within the inflamed tissue [13]. In addition, significantly more lymphocytes expressing TH1-associated receptors CXCR3, CCR5, IL-12R, and IL-18R were observed in sarcoid BAL fluid compared to peripheral blood of sarcoidosis patients, indicating translocation of TH1 cells from blood into the lungs [13]. Taken together, the data suggests that secretion of CXCR3 ligands by lung macrophages is responsible, at least in part, for attracting TH1 cells towards the inflamed lungs [13]. Importantly, elevated CXCL9 and CXCL10 levels in BAL fluid correlated with multisystem organ involvement and with worsening pulmonary function test results, respectively [13]. More recently, a soluble form of the IL-2 receptor (sIL-2R) which is secreted by T-cells upon activation, has shown promise as a more sensitive and specific marker for sarcoidosis compared to traditional tests such as ACE and lysozyme [14,15]. Accordingly, the presence of TH1-associated chemokines, cytokines, and their cognate receptors in chronically inflamed lung tissues not only indicate the presence of sarcoidosis, but could help to explain the pathogenesis of sarcoid granulomas.

In addition, alternatively activated (M2) macrophages and Type 2 CD4^+^ T Helper (TH2) cells can be found in the sarcoid lung [4,16]. These cells secrete IL-10, a suppressive cytokine that typically inhibits TH1 cell responses and counteracts the actions of inflammatory cytokines [16]. Although acute sarcoidosis is generally characterized by a proinflammatory TH1/M1 cell response, increased IL-10 messenger RNA (mRNA) levels have been detected in newly diagnosed patients with active sarcoidosis [17,18]. The indications that these two opposing subtypes of CD4^+^ T-cells and macrophages coexist seem counter-intuitive. However, this could be explained by the assumption that the anti-inflammatory TH2/M2 response balances out the proinflammatory TH1/M1 response in an attempt to keep the overactive TH1/M1 response in check [16]. This balance between the two functionally distinct cell populations may further determine clinical heterogeneity, disease stages, and outcomes as discussed in the following section [19].

## 4. TH1/TH2 Biomarkers and Patient Prognosis

About two-thirds of patients diagnosed with acute sarcoidosis enter remission with good long-term outcomes, while one-third go on to develop chronic sarcoidosis [17].

Patients who recover from sarcoidosis demonstrate a functionally restored TH1 cell response, while those with active disease have abnormalities in intracellular signaling resulting in/indicative of a defective TH1 cell response. In one study, peripheral CD4^+^ T-cells from sarcoidosis patients were found to have spontaneous secretion of IL-2 and IFN-γ that exceeded that of healthy controls [20]. However, upon polyclonal TCR stimulation of both BAL fluid and systemic CD4^+^ T-cells, IL-2, and IFN-γ production became significantly decreased [20]. Additionally, the expression levels of important IL-2 transcriptional mediators such as lymphocyte-specific tyrosine kinase (Lck), protein kinase C-theta (PKC-θ), and Nuclear Factor kappa B (NF-κB) became significantly reduced upon TCR stimulation [20]. Both BAL fluid and systemic CD4^+^ T-cells also displayed reduced proliferative capacity to polyclonal stimulation [20]. These collective findings indicate a state of CD4^+^ T-cell anergy in sarcoidosis. Moreover, patients who spontaneously recovered from sarcoidosis exhibited normalized levels of Lck, PKC-θ, and NF-κB as well as proper response to TCR stimulation by secretion of IL-2 and IFN-γ that approached healthy control levels [20].

In a similar study, NF-κB-p65 subunit deficiencies were more common in patients with stage IV sarcoidosis, multiple organ involvement, and more active disease [5]. NF-κB-p65 deficient sarcoid T-cells showed signs of exhaustion usually caused by persistent antigenic stimulation [5,21]. One such sign is increased expression of inhibitory receptor Cytotoxic T Lymphocyte Antigen-4 (CTLA-4) [5,21]. NF-κB-p65 deficient sarcoid T-cells also exhibited reduced expression of CD3ζ, a chain of the CD3 coreceptor which couples antigen recognition of the TCR to intracellular signal transduction pathways, and expressed low levels of Nuclear Factor of Activated T-cells (NF-AT) which is necessary for CD40 Ligand (CD40L) expression [5,22]. CD40L is a molecule primarily expressed on activated CD4+ T-cells which induce IL-12 production by macrophages as well as TH1 cell differentiation [23]. These findings suggest that the CD4^+^ T-cell signaling defects of sarcoidosis patients results in a functionally impaired TH1 cell response associated with worse disease outcomes.

The TH1 CD4^+^ T-cell dysfunction could be caused by abnormal TCR signaling, absence of co-stimulation and/or overabundance of inhibitory molecules. In fact, higher percentages of CD4^+^ T-cells expressing Programmed Death-1 (PD-1), an inhibitory receptor that negatively regulates proliferation, cytokine secretion and survival of CD4^+^ T-cells, were found in BAL fluid and peripheral blood of patients with sarcoidosis [24]. The sarcoid PD1^+^ CD4^+^ T-cells demonstrated reduced proliferative capacity which was restored upon antibody-mediated blockade of PD-1 [24]. Further, PD-1 expression on CD4^+^ T-cells significantly declined in patients with spontaneous remission of disease compared to patients with active disease [24]. However, whether PD-1 upregulation in sarcoid CD4^+^ T-cells is responsible for the anergic state of these lymphocytes is controversial since other studies have reported different findings [5].

Additionally, a shift towards a TH2 type immune response may be associated with worse sarcoidosis outcomes. Thymus-and-Activation-Regulated Chemokine (TARC), a chemotactic cytokine that binds to C-C Motif Chemokine Receptor type 4 (CCR4), a molecule selectively expressed on effector TH2 CD4^+^ T-cells, is essential to the TH2 cell response since it attracts TH2 cells to sites of allergic inflammation [2,25]. TARC levels as well as the percentage of CCR4^+^ cells were higher in sarcoidosis patients with greater organ involvement and more advanced disease stages compared to patients with less severe disease [26]. Although we do not know if TH2 cell dominance is a cause or a correlation of more severe disease, biomarkers of TH2-driven inflammation are likely to serve useful as indicators of a poor disease prognosis in sarcoidosis. It is possible that TH2 cell response becomes upregulated upon failure or anergy of TH1 cells. A potential TH2 cell marker yet to be studied in relation to sarcoidosis outcomes is GATA-binding protein-3 (GATA-3), the key transcription factor for TH2 cell differentiation [27].

Additionally, immunosuppression seems to be negatively associated with spontaneous remission of acute sarcoidosis, and patients treated with glucocorticoids, the most commonly used drugs for the treatment of pulmonary sarcoidosis, show higher levels of Immunoglobulin (Ig)G4 and IgE which are indicative of a predominant TH2 cell response [20,28]. NF-κB, a transcription factor that induces the expression of various cytokines including IL-12, is downregulated by glucocorticoids, and this may dampen the functional TH1 cell response associated with recovery, possibly skewing the CD4^+^ T-cell response towards a TH2 profile associated with worse outcomes [20]. NF-κB screening could also be used as a tool to rule out glucocorticoids as a therapy option for certain sarcoidosis patients. This is because patients with low starting levels of NF-κB would most likely not benefit from corticosteroids since lowering NF-κB is a major mechanism of action of these drugs [5]. Thus, if NF-κB levels were already low at baseline, administering corticosteroids would have minimal effect since the levels of NF-κB are too low for these drugs to be efficacious. In addition, the deleterious side effects associated with long-term glucocorticoid administration could be altogether avoided for patients with low baseline NF-κB levels.

Patients with chronic sarcoidosis are more likely to eventually develop the deadly complication of lung fibrosis, which may be the result of a predominant TH2 cell response to presumptive sarcoid antigenic stimulation [17,29]. Cytokines such as IL-4 and IL-13 released by TH2 cells stimulate M2 macrophages, which in turn promote activation and proliferation of fibroblasts [30,31]. In fact, chronic sarcoid granulomas stain with higher intensity for the monocyte-macrophage lineage receptor CD163, a marker for M2 macrophage polarization, compared to tuberculous granulomas [4]. Although studies evaluating the relationship between TH2 cell cytokines, most notably IL-13, and fibrotic sarcoidosis are unclear, C-C Motif Chemokine Ligand 18 (CCL18)—a marker of M2 macrophage activation that directly stimulates collagen production in fibroblasts—was found to be significantly elevated in BAL fluid of patients with sarcoidosis-associated pulmonary fibrosis [32,33]. Thus, although TH2 cell cytokines may not be reliable biomarkers of sarcoid fibrosis, markers of alternatively activated macrophages may prove useful as indicators of fibrosing disease in sarcoidosis.

Due to their relative novelty, exact specificity and sensitivity for sarcoidosis has yet to be determined for the majority of the aforementioned TH1/TH2 sarcoid biomarkers. Thus, we suggest that a future diagnostic approach include a panel of some of the discussed biomarkers as a supplemental clinical decision making tool in sarcoid diagnosis and determination of prognosis.

## 5. Serum Amyloid A and Chitotriosidase: General Inflammation and Sarcoidosis Prognosis Markers

Serum Amyloid A (SAA) is an acute phase reactant and nonspecific inflammatory marker synthesized by the liver. It regulates a host of immune cell functions such as leukocyte activation chemotaxis and cytokine production [34]. SAA plasma levels were significantly increased in sarcoid patients compared to healthy controls, and its levels inversely correlated with forced expiratory volume in one second (FEV1), suggesting deteriorating lung function [34]. In fact, sarcoidosis patients with increased SAA plasma levels were more likely to receive higher doses of prednisone than those who had normal levels [34].

SAA was expressed at significantly higher levels in sarcoid granulomas compared to granulomas of other origins and appear to play important regulatory roles in granuloma formation [35]. One of the proposed roles of SAA in adaptive immunity is the regulation of TH cell differentiation. SAA localizes to macrophages and multinucleated giant cells in granulomas and its expression strongly correlated with the number of lung-infiltrating CD3^+^ T lymphocytes [35]. Inside granulomas SAA induces activation of NF-κB through stimulation of Toll-like receptor 2 (TLR-2), a pattern recognition receptor expressed on macrophages and other antigen presenting cell types. This results in TNF-α, IL-10, and IL-18 production by these leukocytes [35]. IL-18 is a pro-inflammatory cytokine that promotes synthesis of IFN-γ, a TH1-promoting cytokine [36]. Thus, SAA could potentially regulate granuloma formation by either augmenting the TH1 response via TNF-α, IL-12 and IL-18 production or reducing the TH1 response via production of IL-10 [35]. If SAA limits the TH1 response, higher levels of SAA may exacerbate the pulmonary disease, which may explain the association between high SAA levels and low FEV1 values. Additionally, SAA has recently been associated to induction of pathogenic Type 17 CD4^+^ T Helper (TH17) cells, the role of which is described in more detail below [37].

Another promising sarcoidosis biomarker is the enzyme chitotriosidase (CTO), which is produced by activated macrophages and polymorphonuclear neutrophils. CTO functions as part of the innate immune response against chitin-covered micro-organisms by breaking down their polysaccharide coatings [38]. Levels of CTO were significantly higher in serum of sarcoidosis patients compared to healthy controls and correlated with severity of disease such as the presence of pulmonary fibrosis, multiorgan involvement, and the need for high-dose corticosteroid therapy [39]. While increased CTO is not specific for sarcoidosis, the enzyme has proven to be a very sensitive biomarker of active sarcoidosis and has more sensitivity and specificity than other commonly used sarcoid biomarkers such as ACE and lysozyme [1,40].

Although the effects of CTO on the adaptive immune system are far from clear, a new study suggested the need for CTO cleavage of chitin to induce TH2-type CD4^+^ T-cells in cryptococcal lung infections [38,41]. A relationship between CTO and TH2-type immune response induction could potentially explain the worse clinical phenotypes observed in sarcoidosis patients who exhibited the highest CTO levels. However, more mechanistic studies aimed to investigate the exact relationship between CTO and TH2 biomarkers are needed to confirm this hypothesis.

## 6. TH17 Biomarkers Play Key Roles in Pathogenesis and Enable Prognosis Prediction

In an interesting twist, recent studies have shown that TH17 CD4^+^ T-cells may contribute equally or more to the pathogenesis of sarcoidosis than the traditionally assumed TH1 CD4^+^ T-cells. TH17 CD4^+^ T-cells are normally involved in recruiting neutrophils to fight bacterial or fungal infections, thus promoting inflammation [42]. TH17 CD4^+^ T-cells have also been shown to cause autoimmunity, and an increased TH17: T regulatory cell ratio has been noted in several autoimmune diseases [42].

Recently, researchers have discovered CD4^+^ T-cells that simultaneously express two chemokine receptors: C-C Motif Chemokine Receptor Type 6 (CCR6), a molecule preferentially expressed by TH17 cells, and CXCR3, a proposed marker for TH1 cells [43]. This new lymphocyte subset called TH1/TH17 or TH17.1 has been found in higher numbers than the classical TH1 CD4^+^ T-cell subtype in BAL fluid of sarcoidosis patients [43]. Like TH1 CD4^+^ T-cells, TH17.1 CD4^+^ T-cells secrete high levels of IFN-γ and may be responsible for most of this cytokine’s secretion in sarcoidosis [43]. TH17.1 CD4^+^ T-cells also secrete proinflammatory cytokine IL-17A, although at much lower levels than the traditional TH17 CD4^+^ T-cell subtype [43]. This information raises the possibility that TH17.1 cells have been misclassified historically as TH1 cells due to their similar cytokine synthesis profile [43].

Interestingly, while the BAL fluid and MLN of sarcoidosis patients contained a greater proportion of TH17.1 CD4^+^ T-cells, the peripheral blood of these patients did not have a prominent TH17.1 cell population but instead, had an increased frequency of TH17 cells [43,44]. It has been suggested that TH17 cells in the circulation exhibit plasticity to differentiate into different TH cell subtypes depending on the local microenvironment [44]. Macrophage-derived IL-23 induced TH17 cells to secrete high IL-17A levels and differentiate into a more pro-inflammatory classical TH17 subtype [45]. TH17 cells could potentially differentiate into the TH17.1 cell subtype in settings of chronic inflammation [44]. In fact, TH17 cells express high levels of CXCR3, a chemokine receptor that helps attract these cells to sites of IFN-γ-mediated inflammation [46]. Additionally, cytokines such as IL-12, seem to drive TH17 cells to differentiate into TH17.1 and even TH1 subtype cells both characterized by high levels of IFN-γ secretion [47]. TH17 and TH17.1 cells also shared TCR-variable β-chain clonotypes which suggests a common clonal origin [47]. Taking this evidence together with the observation that TH17.1 cells predominated in areas of granulomatous inflammation (i.e., BAL fluid and MLN), while TH17 cells are more commonly observed in peripheral blood, it can be deduced that TH17.1 cells arise from circulating TH17 cells when the later enter the inflamed lungs and become exposed to high IL-12 and/or IFN-γ concentrations [43]. Furthermore, peripheral blood sarcoid CCR4^+^ CXCR3^+^ CCR6^+^ T-cells which represent an intermediate stage between classical TH17 cells and TH17.1 cells expressed lower than normal levels of CTLA-4. Lack of CTLA-4 in these cells may contribute to their increased proliferation in the lungs of sarcoidosis patients [44].

The TH17 phenotype as well as specific cytokine patterns within the areas of granulomatous inflammation has significant prognostic implications for sarcoidosis patients.

High percentages of TH17 CD4^+^ T-cells expressing PD-1 have been observed in the peripheral blood of sarcoidosis patients [48]. These cells displayed reduced proliferative capacity, which suggests an anergic state, and through PD-1 signaling they upregulated the expression of Signal Transducer and Activator of Transcription 3 (STAT3) leading to increased IL-17A and Transforming Growth Factor β1 (TGF-β1) production [48]. TGF-β1 is associated with wound-healing and fibrosis, and unsurprisingly, co-culture of sarcoid PD1^+^ TH17 CD4^+^ T-cells with fibroblasts resulted in increased collagen production by fibroblasts [48]. These findings suggest that dysfunctional TH17 cells secreting TGF-β1 may be involved in the development of fibrotic pulmonary sarcoidosis [48]. Thus, high levels of PD-1, STAT3, and/or TGF-β1 associated with predominance of lung TH17 infiltrates could be used as surrogate markers for the development pulmonary fibrosis in patients with sarcoidosis.

The TH17.1 cell subtype and the TH17.1:TH17 ratio has been noted to be higher in BAL fluid of patients developing chronic sarcoidosis compared to those undergoing resolution of disease [44]. These TH17.1 cells were characterized by the production of high concentrations of IFN-γ and low concentrations of IL-17A [43]. Interestingly, the same patients with chronic sarcoidosis also exhibited a diminished TH1 cell population compared to patients undergoing disease resolution [44].

Patients with Löfgren’s syndrome—an acute form of sarcoidosis associated with favorable outcomes and low fibrotic development—were observed to also have increased numbers of BAL fluid TH1/TH17 hybrid cells compared to those with non Löfgren’s syndrome sarcoidosis [49]. The TH1/TH17 hybrid cells were detected by concurrent expression of CCR6 and CXCR3 and of T-bet and RORγT. T-bet and RORγT are the master transcription factors for TH1 and TH17 cell differentiation, respectively [49]. However, Löfgren’s syndrome patients differed from non-Löfgren’s syndrome sarcoidosis patients in that their TH1/TH17 cells secreted significantly lower levels of IFN-γ and significantly higher levels of IL-17A [49].

The evidence suggests that a continuum exists of TH17 cells associated with various cytokine profiles. TH17 cells secreting TGF-β1 are associated with pulmonary fibrosis [48]. In addition, TH17.1 cells with a more TH1 cytokine profile—i.e., IFN-γ expression—are associated with worse clinical outcomes, and those with a more classic TH17 cytokine profile—i.e., IL-17A expression—are associated with better outcomes. Perhaps high IFN-γ secreting TH17.1 cells are the immune system’s attempt to compensate for a defective/anergic TH1 cell response, which is also associated with more progressive disease. Thus, while the detection of TH17 or TH17.1 cells themselves cannot be used as specific markers of sarcoidosis outcome, the presence of TH17/TH17.1 cells along with elevated levels of TGF-β1, IFN-γ, or IL-17A may prove to be a valuable indicator of disease severity and progression.

The various sarcoidosis outcomes associated with the distinct TH17.1 cytokine profiles are potentially due to the presence (or lack thereof) of excessive neutrophil recruitment. TH17 cells indirectly recruit neutrophils through secretion of IL-17A [50]. They can also directly recruit neutrophils through production of chemokine C-X-C motif ligand 8 (CXCL8), a neutrophil chemoattractant [50]. Neutrophil to lymphocyte ratio (NLR) in peripheral blood and neutrophil numbers in BAL fluid increased in proportion to radiological stages/severity of sarcoidosis, while NLR was significantly higher in sarcoidosis patients who developed pulmonary hypertension, a feared complication that associates with poor prognosis [51,52,53]. Thus, NLR and neutrophil numbers may be used as indicators of disease severity and prognosis in sarcoidosis. In fact, increased levels of CXCL8 have been detected in BAL fluid of sarcoidosis stage III patients compared to levels of CXCL8 in Löfgren’s syndrome patients, but further studies are needed to establish a direct cause-and-effect relationship between CXCL8 levels and the extent of neutrophil recruitment to the lungs of patients with pulmonary sarcoidosis [54]. Furthermore, excessive neutrophil degranulation in the lungs may also explain why lung infiltrates and tissue destruction correlated with worse clinical phenotypes.

The numbers of TH17.1-like cells also have implications for treatment. One study analyzing intestinal tissue of Crohn’s Disease patients identified a type of P-glycoprotein expressing/multi-drug resistant (MDR) TH17 cell that produces TH1 and TH17 cytokines upon TCR stimulation [55]. These cells were characterized by high levels of IL-23R expression and a robust response to IL-23 stimulation [55]. This subset of TH17.1 cells were resistant to glucocorticoid therapy and selected for within a mixed T-cell population when exposed to glucocorticoids [55]. Since no such studies of MDR TH17 cells in sarcoidosis exist, investigation of P-glycoprotein expressing TH17 cells is warranted since MDR TH17 cells could indicate ineffective glucocorticoid therapy in patients with this cell subset and thus be potentially used as a clinical treatment decision making tool in sarcoidosis patients.

In summary, TH17 plasticity seems to play a major role in the formation/progression of granulomas through cytokine production, while specific cytokine profiles of TH17 subtypes characterize certain disease outcomes of sarcoidosis. Some of the ways TH17 cytokines seem to influence prognosis include fibrosis promotion and neutrophil attraction. However, it remains to be seen if P-glycoprotein expressing TH17 cells drive the pathogenesis of treatment resistant sarcoidosis.

## 7. T Regulatory Cell Dysfunction May Drive Sarcoidosis Progression

T regulatory cells (Treg), a CD4^+^ T-cell subset that acts to suppress other immune cells from reacting to self or foreign antigens, were found to be increased in the peripheral blood of sarcoidosis patients but present in normal levels in sarcoid BAL fluid [56]. These increased levels of peripheral Treg cells are most significant in patients who go on to develop chronic disease, while those who undergo spontaneous resolution did not have increased Treg cell levels [56]. Although the reason for Treg cell elevation in sarcoidosis and especially chronic sarcoidosis is unclear, Treg cell elevation in peripheral blood may be due to impaired attraction or migration of Treg cells to sites of granulomas. Alternatively, this could also be due to an excessive proliferative response to granulomatous inflammation that local Treg cells failed to control [56]. Both possible explanations present scenarios that could result in decreased self-tolerance and greater susceptibility to chronic disease [56]. Thus, levels of Treg cells in peripheral blood could be used as a biomarker to monitor disease progression. Treg are identified by being CD4^+^ and by expressing high levels of CD25 and Forkhead box P3 (FoxP3), the master transcription factor of Treg cell differentiation [56]. While CD25 is part of the IL-2 receptor and thus not specific to Treg, Treg uniquely express CD25 at high levels to bind/sequester available IL-2 as a means of suppressing other T-cells [56].

The Treg present in sarcoidosis patients have also been shown to be dysfunctional which is linked to worse outcomes [20,57]. Sarcoid Treg demonstrated a reduced ability to suppress other CD4^+^ T-cells and exhibited decreased suppressive cytokine production [20,56]. This decreased suppressive ability of sarcoid Treg may be explained upon examination of cell surface proteins associated with Treg cell function and viability.

Apoptosis-prone Treg cells may explain the dysfunction of Treg with sarcoidosis. Expression of CD95—a pro-apoptotic/death receptor—was increased in sarcoid Treg cells which showed augmented sensitivity toward apoptosis mediated by CD95 engagement, and chronic sarcoidosis patients exhibited higher levels of CD95 on peripheral blood and BAL fluid Treg compared to patients undergoing disease resolution [56,57]. Additionally, TLR-2, a transmembrane cell-surface receptor which can induce apoptosis, was upregulated and exhibited signaling alterations in sarcoidosis peripheral blood Treg [58].

Compared to normal controls, sarcoid Treg expressed significantly higher levels of CTLA-4 in peripheral blood, but significantly lower levels of CTLA-4 in MLN [56,59]. Although CTLA-4 expression on other T-cell subsets is inhibitory, surface CTLA-4 is a major mechanism by which Treg inhibit other lymphocytes and is thus a marker of Treg cell function. Consequently, decreased levels of CTLA-4 expression in MLN Treg suggests impaired Treg cell function when these lymphocytes enter the chronically inflamed lung microenvironment. Additionally, the combined deficiency of CTLA-4 on both TH17 cells and Treg cells may allow further activation of TH17 or TH17.1 cells while hindering Treg cell suppression, possibly explaining the increased TH17:Treg ratio found in sarcoidosis [56]. Interestingly, an increased TH17:Treg ratio in peripheral blood is associated with sarcoidosis relapses after discontinuing steroid therapy, while a decreased TH17:Treg ratio in peripheral blood signaled remission [60]. Monitoring the Treg:TH17 ratio has been proposed as a more sensitive test than ACE or pulmonary function tests to evaluate the possibility of relapses post corticosteroid therapy [60].

The relationship between Treg and TH17 CD4^+^ T-cells is further complicated by the discovery that TH17 cells with a more TH1 cytokine secretion profile—i.e., higher levels of IFN-γ—are resistant to suppression by Treg, while the classical TH1 and TH17 CD4^+^ T-cell subtypes are suppressible by Treg [61]. Thus, even if sarcoidosis Treg remain functional, the development of Treg resistant, IFN-γ-secreting TH17.1 cells may lead to an unchecked lung inflammatory response and increased TH17:Treg ratio in patients with worse clinical outcomes after steroid discontinuation in sarcoidosis [60].

Furthermore, patients who underwent spontaneous resolution were found to have Treg with restored suppressive ability [20]. Additionally, Löfgren’s syndrome patients exhibited Treg expressing higher levels of Inducible Co-Stimulator (ICOS) compared to non-Löfgren’s syndrome sarcoidosis [62]. ICOS is a lymphocyte membrane protein that preferentially promotes IL-10 production thus increasing effectiveness of Treg cell functions [63]. These observations suggest that an intact Treg response to resolve inflammation associated with sarcoidosis is essential for recovery, while dysfunctional Treg fail to suppress sarcoid Ag-specific and/or autoreactive responses that drive the pathology of pulmonary sarcoidosis.

## 8. Homeostatic Imbalance and Dysfunction of Natural Killer T Cells in Sarcoidosis

Natural killer T (NKT) cells are a subset of lymphocytes that express features of both T-cells and natural killer (NK) cells and co-express TCRs together with markers associated with NK cells, such as CD56 and/or CD161 [64]. Functionally, these lymphocytes regulate both innate and adaptive immune responses by their ability to secrete TH1, TH2, and/or TH17 cytokines such as IFN-γ, TNF-α, IL-4, IL-10, IL-13, and/or IL-17 [65].

In sarcoidosis, various, although somewhat inconsistent, reports on NKT cell abnormalities have been described. One study detailed significantly decreased NKT cell frequency in BAL fluid of sarcoidosis patients compared to healthy controls which correlated with increased lymphocytosis of mostly CD4^+^ T-cells in BAL fluid, suggesting resulting amplification of the adaptive immune response [64]. Another study noted no difference in the number of NKT cells in BAL fluid but elevated numbers of NKT cells in the peripheral blood of sarcoidosis patients compared to healthy controls. Moreover, the NKT cells in the BAL fluid of sarcoidosis patients expressed higher levels of inhibitory receptor CD94/NKG2A and lower levels of costimulatory molecule killer cell immunoglobulin-like receptor (KIR) than those of healthy controls. Upon in vitro stimulation, these sarcoid BAL fluid NKT cells also expressed higher levels of IFN-γ and TNF-α than sarcoid peripheral blood NKT cells [66].

The observations of decreased NKT cell frequencies in BAL fluid in one study and increased NKT cell numbers in the periphery in another study suggest impaired attraction or migration of NKT cells to sites of granulomas or an excessive proliferative response to granulomatous inflammation that local NKT cells failed to control, similar to Treg homeostatic abnormalities found in patients with sarcoidosis. However, the increased inhibitory receptor and decreased costimulatory receptor levels expressed by BAL NKT cells suggest a dysfunction in the NKT cell population within sarcoid granulomas such as anergy or exhaustion. In addition, the exuberant cytokine response of BAL fluid NKT cells upon in vitro stimulation suggests inadequate pre-existing activation, also related to anergy. Further studies are needed to determine the exact role of NKT cells in sarcoidosis, but the data presented here suggests that dysfunction of NKT cells in/around granulomas could play a critical role in the pathogenesis of sarcoidosis.

## 9. Abnormalities of the CD8^+^ T Cell Population in Sarcoidosis

Although sarcoidosis is considered a predominantly CD4^+^ T-cell driven disease with elevated CD4:CD8 T-cell ratio, increased CD8^+^ T lymphocytes have been detected in the peripheral blood and lung granulomas of sarcoidosis patients while CD8^+^ T lymphocytes were present at normal levels in sarcoidosis BAL fluid [4,67].

Moreover, sarcoid CD8^+^ T-cells displayed signs of functional abnormality and inactivation. One study noted elevated expression of perforin and granzyme by CD8^+^ T-cells in BAL fluid and peripheral blood of non-Löfgren’s sarcoidosis patients compared to healthy controls and Löfgren’s syndrome patients [67]. Perforin and granzyme are two proteins released by CD8^+^ T-cells to mediate target cell lysis and elevated levels of these two cytotoxic mediators may efficiently kill cells to release autoantigens leading to worsening granulomatous inflammation [2,67].

Although non-Löfgren’s sarcoidosis patients at first showed no difference between healthy controls and Löfgren’s syndrome patients in CD8^+^ T-cell-mediated lysis of target cells, after pre-stimulation with the T-cell-activating lectin phytohemagglutinin A and the mitogenic cytokine IL-2, CD8^+^ T lymphocytes from non-Löfgren’s syndrome sarcoidosis patients exhibited significantly higher target cell lysis rates compared to pre-stimulated CD8^+^ T lymphocytes in healthy controls and Löfgren’s syndrome patients [67,68,69]. Another study noted that diagnosing sarcoidosis based on BAL fluid cell populations is much more accurate when the expression of the activation marker Human Leukocyte Antigen (HLA)-DR on CD8^+^ T-cells is additionally measured. The study showed that CD8^+^ T-cells in sarcoidosis patients displayed significantly lower HLA-DR levels compared to CD8^+^ T-cells in patients with other diffuse parenchymal lung diseases, making determination of HLA-DR expression in CD8^+^ T-cells a good way to distinguish sarcoidosis from other interstitial lung diseases [70,71]. The CD8^+^ T-cell population of sarcoidosis patients is comprised of fewer CD45RA^+^ CCR7^+^ naïve and more CD27^−^ CD28^−^ terminally differentiated effector cells [72]. Additionally, CD8^+^ CD27^−^ lymphocytes were elevated in patients with greater disease severity and need for oral corticosteroid therapy. Thus, measuring CD8^+^ CD27^−^ lymphocyte levels has been proposed as a marker of parenchymal infiltration in sarcoidosis as well as need for future treatment [73].

The above studies describe elevated levels of mostly terminally differentiated memory CD8^+^ T lymphocytes in peripheral blood of patients with severe forms of sarcoidosis which could be a result of inadequate CD8^+^ T-cell stimulation. Chronic stimulation by persistent foreign antigens or autoantigens lead to T-cell exhaustion, and the dominance of CD27^−^ lymphocytes among sarcoidosis effector CD8^+^ T-cells supports this notion since CD27 expression on T-cells is lost upon continual antigenic stimulation [5]. An anergic/abnormal TH1 cell response—also correlated with poor sarcoidosis outcomes—could explain inadequate CD8^+^ T-cell stimulation as well, since a normal TH1 cell response is necessary to stimulate CD8^+^ T-cells through production of IL-2 [5]. Because non-Löfgren’s syndrome sarcoidosis patients only exhibited a higher CD8^+^ T-cell-mediated lysis rate upon pre-stimulation by IL-2, it can be assumed that these sarcoid CD8^+^ T-cells lacked IL-2 beforehand—supporting the concept that inadequate TH1 cell stimulation is the culprit [67]. In summary, the above evidence of CD8^+^ T-cell abnormalities points to an anergic CD8^+^ T-cell compartment in sarcoidosis that results in several detectable biomarkers indicating disease presence, severity, and treatment strategies.

## 10. B Cell Dysfunction and Population Imbalance in Sarcoidosis

Although granuloma formation in sarcoidosis is normally considered T-cell mediated, B cell involvement has been shown to have more influence than previously thought. Increased numbers of B lymphocytes and their progeny, plasma cells, have been observed in areas surrounding granulomas [74]. A recent study noted a higher frequency of peripheral blood B cells in patients with Stage II sarcoidosis compared to healthy controls but noted a reduction in CD19^+^ CD27^+^ IgD^−^ class switched and CD19^+^ CD27^+^ IgD^+^ unswitched memory B cell frequencies in active sarcoidosis [75]. Furthermore, the frequency of activated CD19^+^ IgD^+^ CD38^+^ naïve B cells rose in sarcoidosis compared to inactive disease or healthy controls [75].

The same study noted a change in the T follicular helper (Tfh) cell population which are the CD4^+^ T-cells responsible for helper T-cell-dependent antibody responses [75]. Tfh cells provide contact-dependent and cytokine signals to B cells within germinal centers of lymph node follicles and are identified by high expression of C-X-C Chemokine Receptor Type 5 (CXCR5), a cell membrane receptor necessary for the migration of Tfh cells to B cell-rich lymph node follicles [75]. CXCR3^−^ CCR6^−^ Tfh2 and CXCR3^−^ CCR6^+^ Tfh17 cells increased, while CXCR3^+^ CCR6^−^ Tfh1 and CXCR3^+^ CCR6^+^ CCR4^−^ Tfh17.1-like cells decreased in sarcoidosis patients [75]. It has been shown that Tfh2 and Tfh17 CD4^+^ T-cells can activate and induce IgM, IgG, and IgA secretion in naïve B cells through IL-21 production, while Tfh1 cells can trigger apoptosis in activated naïve B cells [76]. Although the role of Tfh cells in the pathogenesis of sarcoidosis is not well understood, the predominant Tfh subsets may explain the skew in B cells toward an activated naïve phenotype as well as the hypergammaglobulinemia of sarcoidosis [76,77]. Additionally, a Tfh2/Tfh17 dominance has been observed in several other autoimmune diseases [75].

The study also found a significant increase in the percentage of CD5^+^ CD27^−^ and CD24^+++^ CD38^+++^ IL-10 producing regulatory B (Breg) cells [75]. IL-10 is a suppressive cytokine that turn off the TH1 response and thus facilitates non-inflammatory, fibrotic repair. This increase in Breg may be a mechanism to control ongoing granulomatous inflammation or may contribute to the pathogenesis of the disease by amplifying the TH2 response associated with worse clinical outcomes [75]. Another B cell subset known as cytokine-producing B effector cells (Be) may also amplify certain T helper cell responses [78]. Once cultured with either TH1 or TH2 CD4^+^ T-cells in vitro, cytokine-producing Be cells begin producing the characteristic cytokines of the CD4^+^ T-cell subset they were cultured with upon re-stimulation, thus promoting the pre-existing T-cell population’s proliferation and differentiation into effector cells [78]. For example, Be-1 cells cultured with TH1 CD4^+^ T-cells produce TH1-associated cytokines such as IL-12 and IFN-γ, and Be-2 cells cultured with TH2 CD4^+^ T-cells produce TH2-associated cytokines such as IL-4, IL-10, and IL-13 [78]. The relationship between cytokine-producing Be cells and sarcoidosis has yet to be examined but the distinct Be cell subsets may also have important roles in sarcoidosis pathogenesis.

Patients with active chronic sarcoidosis also exhibited significantly elevated levels of B cell-Activating Factor (BAFF), a TNF-ligand family member involved in B cell survival and maturation [77]. Increased BAFF levels in peripheral blood of patients with sarcoidosis negatively correlated with memory B cell frequencies and positively correlated with hypergammaglobulinemia [77]. Another study showed a positive correlation between serum BAFF levels and disease activity and severity and that it may prove to be a useful marker for monitoring sarcoidosis disease outcomes [79]. BAFF is critical to B cell development, and increased BAFF levels may be responsible for the B cell population shifts observed in sarcoidosis or may be part of a compensatory reaction to such B cell population shifts.

Another potential B cell sarcoidosis biomarker is the presence of specific immune complexes (ICs). ICs are formed when B cells produce antibody against antigenic determinants that recognize and bind to the Ag molecules. ICs can circulate in the blood stream, fix complement and deposit in various tissues such as the skin, joints, and kidney. ICs have been detected in the bloodstream of up to 58% of patients with sarcoidosis, but investigation of antigens within the ICs themselves has only begun recently [80]. A study in Japan showed high levels of ICs in sinus macrophages of sarcoidosis patients compared to healthy controls [81]. Upon IHC analysis, 89% of the sarcoid ICs stained positive for *Propionibacterium acnes* bacteria, a pathogen linked to the Japanese variant of sarcoidosis [2,81].

Another recent study done in India showed that up to 70% of sarcoidosis patients had ICs containing *Mycobacterium tuberculosis* antigens. A percentage that is comparable to those contained in sputum smear-negative, culture-positive tuberculosis patients [82]. Thus, ICs antigens give further support to the existing theory of a microbial triggering with sarcoidosis [2]. Furthermore, antigen-specific ICs may present a biomarker candidate for sarcoidosis, although tests to further differentiate between sarcoidosis and tuberculosis are necessary [82].

Upon stimulation by transmembrane cell-surface B cell co-stimulators such as CD40 or TLR9, peripheral blood B cells from severe, chronic sarcoidosis patients exhibited reduced proliferation and expression of activation marker CD25 compared to healthy controls [75,83,84]. The observed anergy of sarcoid B cells may be due to the decreased levels of NF-κB/p65, which are also detected in the B cells of severe chronic sarcoidosis patients [84]. These results show that defects in B cell signaling may be responsible for the B cell dysfunction and population imbalance noted in chronic, persistent sarcoidosis.

Another possible cause of B cell anergy is the lack of co-stimulation from CD4^+^ helper T-cells. Advanced stage sarcoidosis is characterized by anergic CD4^+^ T lymphocytes expressing low levels of NF-ATc2 which is necessary for CD40L and ICOS expression [5,75,85]. CD40L and ICOS expression on helper T-cells play important roles in B cell differentiation, survival, and Ig class-switching [77,85]. If lacking CD40L and ICOS-mediated signaling, B cells could become anergic/dysfunctional and lose the ability to undergo isotype switching, which may explain the deficit of memory class-switched B cells observed in sarcoidosis [5,77]. In further support of this idea, the deficit in sarcoidosis peripheral blood memory B cells has been noted to be mostly due to a decrease in CD27^+^ IgM^+^, CD27^+^ IgG^+^, or CD27^+^ IgA^+^ T-cell-dependent B memory cells, while CD27^−^ IgA^+^ T-cell-independent B cells increased [74]. However, it is unknown if the decrease in blood memory T-cell-dependent B cells is due to localization of these cells in and around granulomas or an actual complete deficit. The anergy noted in the T-cells responsible for stimulating B cells and the B cells themselves could also be associated with continual antigenic stimulation by persistent sarcoid antigens. The resulting memory cell deficit may contribute to less class-switched, high-affinity memory antibody responses thus compounding the problem of decreased antigen clearance. The above observations lend supportive evidence to the importance of an effective humoral response in sarcoidosis.

## 11. Activation of Intracellular Signaling Pathways and Molecular Biomarkers in Sarcoidosis

Molecular markers further confirm the above observations on the adaptive immune system’s role in sarcoidosis pathogenesis as well as give further insight to associated intracellular signaling pathways.

When compared to BAL cells of healthy individuals, microarray analysis of sarcoid BAL cells shows unique transcriptional profiles with distinct upregulated pathways most notably in the adaptive immune system, although the cytotoxicity pathway of NK cells is also upregulated. These include the TH1 associated IFN-γ and IL-12 signaling pathways and the TH17 associated IL-17 and Il-23 signaling pathway. Interestingly, members of the proteosome pathway were also upregulated in sarcoidosis patients compared to healthy controls. The proteosome is known to be involved in Class I major histocompatibility complex (MHC) presentation and inflammatory response regulation through NF-κB activation leading to expression of the TH1 and TH17-associated cytokines TNF-α, IL-1, and IL-8. This newly discovered link between pro-inflammatory response, adaptive immunity and proteosome pathways warrants further investigation and presents possible novel treatment targets involving the proteosome such as drugs like bortezomib [86].

The Janus kinase (JAK)-STAT pathways have also been shown to be active in sarcoidosis, especially in the TH1 and TH17 CD4^+^ T-cell subtypes. IL-6, a cytokine that activates the main TH17 transcription factor STAT3, has significantly increased mRNA levels in sarcoid granulomas compared to suture granulomas, while IFN-γ, a TH1 cytokine induced by STAT4 and which activates STAT1, has increased mRNA levels in sarcoid granulomas compared to suture and fungal granulomas [87]. Furthermore, RNA sequencing has shown significantly higher STAT1 and STAT3 levels in cutaneous sarcoidosis compared to healthy controls and patients with xanthelasma, a different granulomatous cutaneous disease [88]. The involvement of JAK-STAT pathways also presents promising treatment options, and treatment with Tofacitinib, a JAK-STAT inhibitor, has been shown to aid in resolution of cutaneous sarcoidosis characterized by concurrent down-regulation of IL-6 and IFN-γ mRNAs [88].

Another major pathway implicated in sarcoidosis is the mammalian target of rapamycin (mTOR) pathway which plays a critical role in cell nutrition and immune responsiveness [89]. A next generation sequencing study of five French families with familial sarcoidosis revealed inherited pathogenic variants of genes encoding mammalian target of rapamycin complex 1 (mTORC1) and mTOR complex [90]. Additionally, a similar Chinese study on one family with three members diagnosed with sarcoidosis reported mutations in the monocyte chemoattractant protein-1 gene (*MCPIP1*), a transcription factor that works through mTORC1 signaling, resulting in negative effects on TH17 differentiation—further linking TH17 cell dysfunction to sarcoidosis pathogenesis [91]. Thus, mTOR inhibitors, such as sirolimus, should be further investigated as a potential therapy option for sarcoidosis.

## 12. Conclusions

Given the unique utility of individual biomarkers discussed, the most useful and accurate approach to simultaneously diagnose, stage, and predict prognosis in sarcoidosis may be to measure several biomarkers in a comprehensive blood or BAL fluid panel. However, a comprehensive history and physical exam remains indispensable in diagnosing sarcoidosis in addition to ruling out other pulmonary diseases. For example, a thorough social history will help exclude other causes of lymphadenopathy/lung granulomas caused by occupational exposures such as berylliosis and silicosis, and index of suspicion for malignancy will be influenced by smoking history. If clinical suspicion is high, infectious and malignant causes of hilar lymphadenopathy must also be ruled out using appropriate testing, such as sputum culture for tuberculosis.

Due to the less invasive nature of blood tests, we recommend that a serum test panel be used initially if sarcoidosis is suspected. Serum tests that show promise for detection of sarcoidosis include serum CTO and sIL-2R which have higher sensitivity and specificity than the more commonly used sarcoid biomarkers ACE and lysozyme [14,15,40]. Once sarcoidosis-specific biomarker parameters are determined relative to other conditions, flow cytometry of peripheral blood samples could be used to look for a high percentage of TH1 and TH17 CD4^+^ cells expressing PD-1, high frequency of TH17 and TH17/TH17.1 intermediates, and high levels of Treg with increased CD95/CTLA-4 expression—markers of T-cell dysfunction in sarcoidosis [24,44,48,56].

BAL and IHC staining of lung biopsies will also be useful in differentiating between sarcoidosis and other granulomatous pulmonary diseases. A significantly increased CD4:CD8 T-cell ratio makes sarcoidosis more likely than other interstitial lung diseases [6]. Additionally, IHC of lung biopsies have shown that SAA levels in sarcoidosis granulomas significantly exceed that of granulomas from berylliosis, infectious causes such as histoplasmosis or tuberculosis, hypersensitivity pneumonitis, Wegener granulomatosis, pulmonary malignancy, and other inflammatory lung diseases [35].

Once sarcoidosis is diagnosed, we recommend monitoring of disease progression by measuring baseline levels of certain biomarkers and following their levels over time. For example, a simultaneously increased BAL fluid level of CCL18 and TGF-β1 secreting CCR6^+^ CD4^+^ T-cells would raise clinical suspicion of fibrosing sarcoidosis, and the clinician could consider prescribing anti-fibrotic agents to minimize the damage of future fibrotic scarring. Thus, in the future, the use of more specific biomarkers could allow for individualized, patient-centered management of sarcoidosis to yield better patient outcomes (Table 1).

The various cell types involved in sarcoidosis not only allow for multiple methods of detecting and monitoring disease but also point to treatment targets as well. The effectiveness of medications that inhibit CD4^+^ T-cell function such as corticosteroids and antimalarials show the importance of T helper cells in the pathogenesis of sarcoidosis [92]. However, while corticosteroids cause lymphocytosis, they have a broad range of other immune and nonimmune cell targets as well and only offer short-term relief of symptoms [92]. Furthermore, antimalarials can cause serious side effects [93]. Thus, although medications that inhibit pathogenic CD4^+^ T-cells show some clinical benefit in sarcoidosis, a recent review suggested that the goal of future treatments should be to ultimately restore T-cell function since re-invigorated T-cell function corresponded with clinical resolution [20,92].

Therapy targeting the B cell specific marker CD20 with Rituximab, an anti-CD20 monoclonal antibody, has been effective in resolving cases of sarcoidosis refractory to corticosteroid therapy [94]. However, Rituximab has also been shown to trigger sarcoidosis-like reactions several months after successfully being used to treat other autoimmune diseases. Rituximab-induced sarcoidosis may be caused by the drug induced imbalance between B cell populations characterized by higher levels of naïve B cells and a deficit in memory B cells. Rituximab-induced ablation of pre-existing B cells triggers de novo production of naïve B cells as the bone marrow repopulates the B cell pool post-treatment [95]. Additionally, Rituximab may also skew the ratio of IL-10 producing Breg cells to classical B cells. Interestingly, successful treatment (i.e., complete remission) with Rituximab of the autoimmune disease pemphigus vulgaris is characterized by a higher number of IL-10 producing Breg post-treatment compared to cases with incomplete remission [96]. The skewed B cell population post-successful Rituximab treatment resembles the B cell population characteristic of sarcoidosis patients and thus may help explain the pathogenesis of sarcoidosis.

Furthermore, the evidence of lymphocyte anergy in sarcoidosis points to chronic stimulation by persistent antigens, superantigens, or autoantigens [5]. Although no antigen has been proven as the exact instigator, multiple candidates have been suggested.

Vimentin is a possible autoantigen proposed as the culprit of the granulomatous inflammation seen in sarcoidosis. IHC staining identified increased vimentin especially localized to granulomas in sarcoidosis patient spleens and stimulation of sarcoid peripheral blood mononuclear cells with vimentin induced higher IFN-γ and TNF-α secretion compared to those of tuberculosis patients or healthy controls [97].

The discovery of bacterial antigen-associated immune complexes, mycobacterial DNA, and even genes encoding DNA gyrase subunit A in sarcoidosis lung specimens provide evidence for microbial involvement in sarcoid pathogenesis as well as the possible usefulness of antibiotics as a therapeutic option [82,98]. Since a strong TH1 and TH17 response have been shown to be protective against M. tuberculosis, an interplay between sarcoidosis and mycobacterium may also explain why a functionally restored TH1 and TH17 response is necessary for sarcoidosis resolution [99]. The mycobacterial antigens may also persist, and cross-react with self-molecules to induce an autoimmune reaction. One possible mycobacterial antigen is *Mycobacterium tuberculosis* catalase–peroxidase (mKatG) because it has been more frequently identified in sarcoidosis tissues in comparison to healthy controls [100]. mKatG antibodies have also been detected in some sarcoidosis patient serum, while the antibodies are absent in healthy controls [100]. Thus, not only can potential antigens like mKatG and vimentin help identify possible disease triggers, but they may also prove useful as additional biomarkers of the disease.

## Figures and Tables

**Figure 1 ijms-21-07398-f001:**
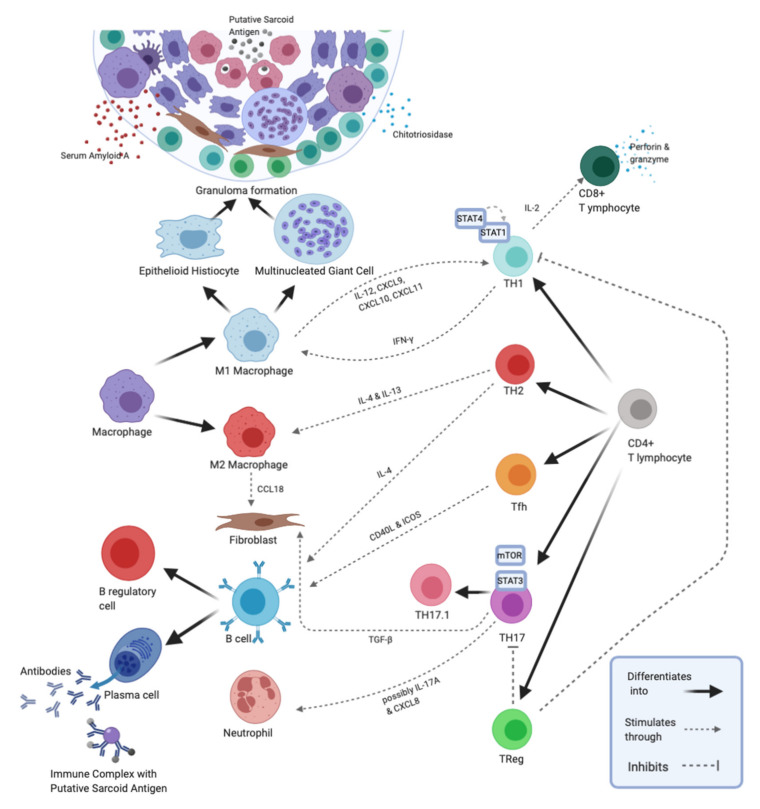
An overview of the serum and bronchoalveolar lavage (BAL) fluid biomarkers produced by cells of the innate and adaptive immune system during granuloma development in sarcoidosis. Granulomas are tightly packaged clusters of cells comprising a central core of macrophages, epithelioid histiocytes, multinucleated giant cells and unknown sarcoid antigens surrounded by a lymphocyte collar. The lymphocyte collar contains, mainly, CD4^+^ T-cells, but CD8^+^ T-cells, Treg cells, B cells, plasma cells and fibroblasts can also be found. During granuloma development, a number of biomarkers are produced and released by these cells. CD4^+^ T lymphocytes are key players in granuloma formation. They differentiate into specific TH cell subtypes (i.e., TH1, TH2, T follicular helper (Tfh), TH17, TH17.1 and Treg) depending mainly on the cytokine microenvironment. The TH1, TH17 and TH17.1 subtypes produce inflammatory markers (i.e., Interferon (IFN)-γ, IL-17A, and IFN-γ/IL-17A, respectively). Through IL-2 production the TH1 subtype helps CD8^+^ T-cells differentiate into cytotoxic effectors which produce biomarkers of inflammation (i.e., perforin and granzyme), while the TH17 subtype attracts neutrophils via IL-17A and C-X-C motif ligand 8 (CXCL8), further contributing to inflammatory marker production. The TH2 and TH17 subtypes can secrete IL-4 (Interleukin-4)/IL-13 and Transforming Growth Factor β1 (TGF-β1), respectively, which are biomarkers of fibrosis. JAK-STAT signaling is strongly implicated in sarcoidosis pathogenesis with Signal Transducer and Activator of Transcription 3 (STAT3) and STAT1/STAT4 playing important roles in TH17 and TH1 cell differentiation, respectively. Altered mammalian target of rapamycin (mTOR) signaling negatively affects TH17 differentiation and production of TH17-associated inflammatory biomarkers. Through expression of CD40 Ligand (CD40L) and Inducible Co-stimulator (ICOS), the Tfh subtype helps B cells differentiate into plasma cells which secrete antibodies to sarcoid antigens. Sarcoid antigens and their specific antibodies form immune complexes that can potentially be used as specific biomarkers of sarcoidosis. The Treg subtype negatively controls production of inflammatory markers by the above-mentioned TH cell subsets. Natural killer T (NKT) cells (not shown) also modulate the CD4^+^ T-cell immune response. Macrophages are also important for granuloma formation and produce different biomarkers depending on their polarization state. M1 macrophages release inflammatory biomarkers (i.e., Serum Amyloid A (SAA), Chitotriosidase (CTO), IL-12, CXCL9, CXCL10, and CXCL11), while M2 macrophages produce biomarkers of fibrosis (i.e., C-C motif ligand 18 (CCL18) and Transforming Growth Factor β1 (TGF-β1)). Image created using BioRender.

**Table 1 ijms-21-07398-t001:** Sarcoidosis biomarkers. Most of the novel biomarkers examined in sarcoidosis have not been clinically validated nor are sufficiently specific or sensitive to be used in isolation for clinical-decision making. However, several sarcoidosis biomarkers have important roles in the clinical management of sarcoidosis when used in combination with clinical data including the results of other biomarkers.

Phenotype	Biomarker	Source/Associated Cell	Source of Sample	References
**Better Outcomes**				
Eventual Remission	Normalized levels of lymphocyte-specific tyrosine kinase (Lck), protein kinase C-theta (PKC-θ), and Nuclear Factor kappa B (NF-κB) expression; normal secretion levels of IL-2 and IFN-γ	TH1 CD4^+^	BALF	[21]
	Decreased Programmed Death-1 (PD-1) on CD4^+^ cells	CD4^+^	BALF and peripheral blood	[25]
	Decreased TH17:T regulatory cell (Treg) ratio	CD4^+^	peripheral blood	[61]
Löfgren’s Syndrome	CCR6^+^CXCR3^+^ cells/T-bet and RORγT co-expressing cells predominantly secreting IL-17A	TH17.1 CD4^+^	BALF	[4]
	High Inducible Co-Stimulator (ICOS) on CD4+ CD25high FoxP3high cells	Treg	BALF	[63]
	Less perforin and granzyme compared to non-LS	CD8^+^	BALF and peripheral blood	[68]
**Worse Outcomes**				
Chronic Sarcoidosis	Increased CD163^+^	M2 Macrophage	lymph node and lung	[4]
	Increased CCR6^+^ CXCR3^+^ cells predominantly secreting IFN-γ	TH17.1 CD4^+^	BALF	[44,45]
	Increased CD4^+^ CD25high FoxP3high cells in peripheral blood	Treg	Peripheral blood	[57]
	Increased CD95 on CD4^+^ CD25high FoxP3high cells	Treg	BALF and peripheral blood	[57]
	Reduced CD25^+^ expression upon stimulation	B cells	Blood	[85]
	Decreased NF-κB-p65 subunit in peripheral B cells	B cells	Blood	[85]
Active/Severe Disease	Increased BAFF secretion	B cells	Blood	[78]
Advanced Radiological Disease Stage	Increased serum TARC and CCR4^+^ cells	TH2 CD4^+^	Peripheral blood	[27]
	High neutrophil:lymphocyte ratio; neutrophil numbers		BALF and peripheral blood	[52,53]
Pulmonary hypertension	High neutrophil:lymphocyte ratio		Peripheral blood	[54]
Systemic organ involvement	CXCL9 levels	Macrophages	Peripheral blood	[16]
	Decreased NF-κB-p65 subunit	CD4^+^	BALF and peripheral blood	[5]
	Increased chitotriosidase	Macrophages	Peripheral blood	[40]
Worse pulmonary function tests	CXCL10 levels	Macrophages	Peripheral blood	[16]
	Increased SAA levels	Liver	Peripheral blood	[35]
Fibrosis	Increased CCL18	M2 Macrophage	BALF	[33]
	Increased chitotriosidase	Macrophages	Peripheral blood	[40]
	Increased CD152; decreased NF-κB-p65 subunit, CD3zeta, and NF-ATc2 in CD4^+^ cells	CD4^+^	BALF and peripheral blood	[5]
	Increased TGF-B secretion and STAT3 and PD-1 expression in TH17 cells	CD4^+^ TH17	Peripheral blood	[49]
**Response to Corticosteroid Therapy**				
Requires increased corticosteroid therapy	High CD8^+^ CD27^−^ cells	CD8^+^ cells	Peripheral blood	[74]
	Increased SAA levels	Liver	Peripheral blood	[35]
	Increased Chitotriosidase	Macrophages	Peripheral blood	[40]
May not benefit from corticosteroids	low NF-κB	CD4^+^ cells	BALF and peripheral blood	[5]
Relapses post corticosteroid therapy	Increased TH17:Treg ratio	CD4^+^	Peripheral blood	[61]

Interleukin-2 (IL-2); Interferon gamma (IFN-γ); Bronchoalveolar lavage fluid (BALF); C-C Chemokine receptor 6 (CCR6); C-X-C Chemokine Receptor 3 (CXCR3); RAR-related Orphan Receptor gamma Thymus-specific isoform (RORγT); B cell-Activating Factor (BAFF); Thymus-and-Activation-Regulated Chemokine (TARC); C-X-C motif ligand 9 (CXCL9); Serum Amyloid A (SAA); Nuclear Factor of Activated T cells (NF-ATc2); Signal Transducer and Activator of Transcription 3 (STAT3).

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
