# Peer review of "Key Players and Biomarkers of the Adaptive Immune System in the Pathogenesis of Sarcoidosis"

_ijms, 2020, doi:10.3390/ijms21197398_

Round 1

Reviewer 1 Report

Re: “Key Players and Biomarkers of the Adaptive Immune System in the Pathogenesis of Sarcoidosis”. Manuscript Revision.

This review proposes an updated review to the biomarkers associated to the Adaptive Immune System in sarcoidosis. The title suggests all of them related to the pathogenesis; however, the manuscript abstract states that “groups distinctive biomarkers can also be used to determine disease progression, predict prognosis, and make treatment decisions”. Compared to therapeutics the diagnosis and detection is barely mentioned in the conclusions; despite the fact that the abstract clearly anticipates that the review is based in immunological biomarkers from the disease progression prognosis and treatment perspectives.

The author of this peer review manuscript “propose that accuracy of diagnosis can be improved if biomarkers of altered lymphocyte populations and levels of signaling molecules involved in disease pathogenesis are measured for patterns unique to sarcoidosis”. However, it is not clear which “patterns” of unique sarcoidosis are they referring to, there is not a section where they explain this categorization stated in the abstract.

The authors present the same TH1/TH2 markers if two different sections, which are very extensive and can be condensed. They may also consider to add a diagram or table that allow the readers to visualize the markers that overlap in both, detection and prognosis sections.   Under the section with the heading “TH1/TH2 Cell Subtype Shifts & Detection of Sarcoidosis” very few emphases if given to the diagnosis specificity of the markers that are derived from this subphenotype. In line 65 the pulmonary diseases compared to sarcoidosis should be mentioned. This section needs more expansion emphasizing the work done for specifically detecting or differentially diagnosing sarcoidosis, in other words how specific are those markers for sarcoidosis only.

Subheading for TH17 on line 628 needs to be more precise, key player is not well representative of the description of this section. On the other hand, prognosis and pathogenesis role of TH17 is mixed in both sections. I suggest incorporating a clear summary at the end of each of these two sections, emphasize what are the authors are trying to accomplish by presenting them by separate.

The authors have missed some immunological molecular biomarkers in sarcoidosis that have been identified by gene expression and epigenetic analysis, only few citations referencing transcription of IL-10 mRNA [16] and IL-2 [18] are mentioned in this context [16]. Expand your review for more immunological targets using microarrays and NGS. For example, multiple STATs (STAT1, STAT3, STAT4) have been identified as dysregulated in sarcoidosis by gene expression.

In regards the implication for treatment suggested in the last paragraph of TH17 Biomarkers: Predictor of Patient prognosis, the authors refer to Chron’s disease and not sarcoidosis. Although granulomatous formation is a common histopathological finding, both diseases are different and further research in needed in sarcoidosis.

Author Response

Thank you for giving us the opportunity to submit a revised version of the manuscript ‘Key players and biomarkers of the adaptive immune system in the pathogenesis of sarcoidosis’ for publication in the International Journal of Molecular Sciences, we appreciate the time and effort that you dedicated to providing feedback on our manuscript, and are grateful for the insightful comments on and valuable improvements to our paper.

We have incorporated most of the suggestions made by you. Those changes are highlighted within the manuscript. Please see below, in blue, for a point-by-point response to your comments and concerns. All page numbers and lines refer to the revised manuscript file with tracked changes.

Reviewer’s Comments to the Authors:

Reviewer 1.

  1. Compared to therapeutics the diagnosis and detection is barely mentioned in the conclusions; despite the fact that the abstract clearly anticipates that the review is based in immunological biomarkers from the disease progression prognosis and treatment perspectives.

Thank you for pointing this out. The reviewer is correct and we have discussed  this in more detail in the conclusions. Diagnosis of sarcoidosis is made by exclusion of competing diagnosis and so far there is no test specific for this disease. Thus, a comprehensive history, physical exam and multiple testing remains indispensable in diagnosing the sarcoidosis. We recommend that in addition to the above, test panels including combination of biomarkers should be run to help with diagnosis, particularly when the clinical-radiological features at presentation suggest an equally possible diagnostic alternative. We have given some examples of these panels and differentials that can be found in page 13, lines 607 to 630.

  1. The author of this review manuscript “propose that accuracy of diagnosis can be improved if biomarkers of altered lymphocyte populations and levels of signaling molecules involved in disease pathogenesis are measured for patterns unique to sarcoidosis”. However, it is not clear which “patterns” of unique sarcoidosis are they referring to, there is not a section where they explain this categorization stated in the abstract.

We agree with the reviewer’s comment. As suggested, we have changed (softened) this statement to read ‘patterns suggestive of sarcoidosis’ (page 1, line 12). In addition we have made clear that ‘Due to their relative novelty, exact specificity and sensitivity for sarcoidosis has yet to be determined for the majority of the aforementioned biomarkers. Thus, we suggest that a future diagnostic approach include a panel of some of the discussed biomarkers as a supplemental clinical decision making tool in sarcoid diagnosis and determination of prognosis. Corrections addressing this reviewer’s point can be found in page 6, line 216-219 and in Table 1 caption.

  1. The authors present the same TH1/TH2 markers if two different sections, which are very extensive and can be condensed. They may also consider to add a diagram or table that allow the readers to visualize the markers that overlap in both, detection and prognosis sections. 

While we appreciate the reviewer’s feedback, keeping the two sections on TH1/TH2 will help the reader better understand how a given biomarker represents a pathophysiologic pathway involved in sarcoidosis development.  However, we think that including a diagram to visualize the markers and adding a table as a summary are excellent suggestions. We have included a figure and a table in the revised manuscript. These can be found in pages 2 and 15, respectively. Thank you for pointing this out.

  1. Under the section with the heading ‘TH1/TH2 Cell Subtype Shifts & Detection of Sarcoidosis’ very few emphasis if given to the diagnosis specificity of the markers that are derived from this sub-phenotype.

The reviewer is correct and we have added the following statement: ‘Due to their relative novelty, exact specificity and sensitivity for sarcoidosis has yet to be determined for the majority of the aforementioned TH1/TH2 sarcoid biomarkers. Thus, we suggest that a future diagnostic approach include a panel of some of the discussed biomarkers as a supplemental clinical decision making tool in sarcoid diagnosis and determination of prognosis’. Corrections addressing this reviewer’s point can be found in page 6, line 216-219 and in Table 1 caption.

  1. In line 65 the pulmonary diseases compared to sarcoidosis should be mentioned. This section needs more expansion emphasizing the work done for specifically detecting or differentially diagnosing sarcoidosis, in other words how specific are those markers for sarcoidosis only. 

Thank you for pointing this out. To address this we have included common differentials of sarcoidosis, the use of biomarkers to help differentiate sarcoidosis from other similarly presenting diseases, and the reported specificity of the discussed biomarkers. These corrections can be found in page 3, lines 84-91.

  1.  Subheading for TH17 on line 628 needs to be more precise, key player is not well representative of the description of this section. On the other hand, prognosis and pathogenesis role of TH17 is mixed in both sections. I suggest incorporating a clear summary at the end of each of these two sections, emphasize what are the authors are trying to accomplish by presenting them by separate.

We agree with the reviewer’s comment. As suggested, we have changed the subheading of this section to ‘TH17 Biomarkers Play Key Roles in Pathogenesis & Enable Prognosis Prediction’. This correction can be found on page 6, line 258. Additionally, a summary has been added to the section, which can be found on page 8, line 352-357.

  1.  Expand your review for more immunological targets using microarrays and NGS. For example, multiple STATs (STAT1, STAT3, STAT4) have been identified as dysregulated in sarcoidosis by gene expression.

Thank you for pointing this out. To address the point raised by the reviewer we have written a whole new section to present new sarcoid molecular targets like STAT1, STAT3, STAT4, mTORC1 and proteosome. This section entitled ‘Activation of Intracellular Signaling Pathways & Molecular Biomarkers in Sarcoidosis’ can be found on page 12-13, lines 568-603.

  1. In regards the implication for treatment suggested in the last paragraph of TH17 Biomarkers: Predictor of Patient prognosis, the authors refer to Crohn’s disease and not sarcoidosis. Although granulomatous formation is a common histopathological finding, both diseases are different and further research is needed in sarcoidosis.

We agree with the reviewer’s comment. As suggested, we have changed this statement to make clear that sarcoidosis and Crohn’s are different diseases and that pathophysiological extrapolations have to be taken cautiously. This correction can be found on page 8, lines 347-351.

Reviewer 2 Report

Sarcoidosis is a systemic granulomatous disorder which causes and pathogenesis is still largely unknown. Most of the recent studies are dealing with granulomas and the role of macrophages derived cells. The review “Key Players and Biomarkers of the Adaptive Immune System in the Pathogenesis of Sarcoidosis” by  Zhou and Arce covers  all what is already reported about the various lymphocytic subpopulations in sarcoidosis, from T subpopulations to B cells subpopulations. It is really a very thorough and complete exhaustive work which will be very useful for those interested in sarcoidosis.

I have a “major”  comment:

  • As a conclusion tables should be added summarizing which cells and biomarkers reviewed in the manuscript would be investigated in  blood, BAL and granulomas for clinical purpose ie diagnosis, prognosis or treatment follow-up or indication

 Minor comment: It would have been interesting to add a short paragraph on Natural Killer T cells and sarcoidosis.

Author Response

Thank you for giving us the opportunity to submit a revised version of the manuscript ‘Key players and biomarkers of the adaptive immune system in the pathogenesis of sarcoidosis’ for publication in the International Journal of Molecular Sciences, we appreciate the time and effort that you dedicated to providing feedback on our manuscript, and are grateful for the insightful comments on and valuable improvements to our paper.

We have incorporated all the suggestions made by you. Those changes are highlighted within the manuscript. Please see below, in blue, for a point-by-point response to your comments and concerns. All page numbers and lines refer to the revised manuscript file with tracked changes.

Reviewer’s Comments to the Authors:

Reviewer 2.

  1. This is really a very thorough and complete exhaustive work which will be very useful for those interested in sarcoidosis.

Thank you so much for your kind comment. We really appreciate it.

  1. I have a “major”  comment: As a conclusion tables should be added summarizing which cells and biomarkers reviewed in the manuscript would be investigated in  blood, BAL and granulomas for clinical purpose i.e. diagnosis, prognosis or treatment follow-up or indications.

We appreciate the reviewer’s feedback. We agree that including a diagram to visualize the markers and adding a table as a summary are excellent suggestions. We have included a figure and a table in the revised manuscript. These can be found in pages 2 and 15, respectively. Thank you for pointing this out.

  1. Minor comment: It would have been interesting to add a short paragraph on Natural Killer T cells and sarcoidosis.

Thank you for suggesting to include a paragraph on NK T cells and sarcoidosis. This is promising novel area of investigation worth to be discussed. To address this we have written a whole new section to review what is known on NK T cells and sarcoidosis, and NK T cells biomarkers that could be inferred from this knowledge. This new section entitled ‘Homeostatic Imbalance & Dysfunction of Natural Killer T Cells in Sarcoidosis’ can be found on pages 9-10, lines 415-442.

Round 2

Reviewer 1 Report

I appreciate the authors revisions.  A minor revision is suggested:

-Add references associated to each of the parkers presents on Table1

Author Response

Thank you for giving us the opportunity to submit a revised version of the manuscript ‘Key players and biomarkers of the adaptive immune system in the pathogenesis of sarcoidosis’ for publication in the International Journal of Molecular Sciences, we appreciate the time and effort that you dedicated to providing feedback on our manuscript, and are grateful for the insightful comments.

We have incorporated the suggestion made by you. This change is highlighted in yellow within the manuscript. Please see below, for a point-by-point response to your concern. The page numbers refer to the revised manuscript file with tracked changes.

Reviewer’s Comments to the Authors:

Reviewer 1.

  1. Add references associated to each of the parkers presents on Table1

Thank you for pointing this out. The reviewer is correct and we have added the references to the table as suggested. These can be found in pages 14 and 15.
